# Diamond Deposition on Iron and Steel Substrates: A Review

**DOI:** 10.3390/mi11080719

**Published:** 2020-07-24

**Authors:** Xiaoju Li, Lianlong He, Yuanshi Li, Qiaoqin Yang

**Affiliations:** 1State Key Laboratory of Microbial Technology, Shandong University, Qingdao 266237, China; 2Shenyang National Lab of Materials Science, Institute of Metal Research, University of Chinese Academy of Sciences, Shenyang 110016, China; llhe@imr.ac.cn; 3Department of Mechanical Engineering, University of Saskatchewan, 57 Campus Drive, Saskatoon, SK S7N5A9, Canada; yul088@mail.usask.ca

**Keywords:** CVD diamond, diamond like carbon (DLC), steel, carburization, graphite, interlayers, Al alloying, aluminizing, siliconizing, nucleation

## Abstract

This article presents an overview of the research in chemical vapor deposition (CVD) diamond films on steel substrates. Since the steels are the most commonly used and cost-effective structural materials in modern industry, CVD coating diamond films on steel substrates are extremely important, combining the unique surface properties of diamond with the superior toughness and strength of the core steel substrates, and will open up many new applications in the industry. However, CVD diamond deposition on steel substrates continues to be a persistent problem. We go through the most relevant results of the last two and a half decades, including recent advances in our group. This review discusses the essential reason of the thick catalytic graphite interlayer formed on steel substrates before diamond deposition. The high carbon diffusion in iron would induce severe internal carburization, and then voluminous graphite precipitated from the substrate. In order to hinder the catalytic graphite formation, various methods have been applied for the adherent diamond film deposition, such as pre-imposed various interlayers or multi-interlayers, special controls of the deposition process, the approaches of substrate alloying and so on. We found that adherent diamond films can be directly deposited on Al alloying steel substrates, and then the role of Al alloying element was examined. That is a thin dense amorphous alumina sublayer in situ formed on the alloying substrate, which played a critical role in preventing the formation of graphite phase and consequently enhancing diamond growth and adhesion. The mechanism of Al alloying suggests that the way used to improve hot corrosion resistance is also applicable. Then, some of the hot corrosion resistance methods, such as aluminizing, siliconizing, and so on, which have been used by some researchers examining CVD diamond films on steel substrates, are reviewed. Another way is to prepare diamond-like carbon (DLC) films on steel substrates at low temperature, and then the precipitated graphite from the internal carburization can be effectively avoided. In addition, based on some new findings, the understanding of the diamond nucleation and metastable growth is discussed.

## 1. Introduction

Diamond has many excellent properties and is an ideal material for many applications. Since natural diamond is very rare and expensive, creating artificial diamond has attracted voluminous attention. Until now, the high pressure high temperature (HPHT) technique and the chemical vapor deposition (CVD) technique are two ways to synthesize diamond. Using the HPHT technique makes it possible to fabricate bulk diamond by effecting the phase transition from solid graphite to diamond [1,2,3]. As for using the CVD technique, it is difficult to obtain viable diamond. Nonetheless, the CVD technique makes it possible to provide large-area diamond films on various substrates [4,5,6], and to fabricate diamond films with controlling properties such as free formation, crystal phase, film thickness and doping [7,8]. Therefore, the CVD diamond technique is regarded as an essential technique for the future. After more than 30 years of research and development, diamond films synthesis by means of low pressure CVD became economically important. In particular, CVD diamond films on steel substrates have attracted significant interest. That is because the successful coating can combine the individual advantages of diamond, which are superhard, chemically inert, and wear resistant, with the low cost of steel cores that possess superior mechanical/physical performances. Accordingly, many new applications in the industry will open up [9].

However, CVD diamond films on steel substrates are a great challenge and come with several problems. In catalytic CVD process, carbon nanotubes are produced by the catalytic decomposition of hydrocarbon vapors. Cobalt, iron, nickel and their alloys are the most widely used catalysts in carbon nanotubes production through CVD [10,11]. It is well known that the strong catalytic effect of these transition metals, especially iron, inhibits the diamond formation even in a suitable gas phase environment for diamond growth [12,13]. The surfaces of these metal substrates would turn black in a very short period of time due to the rapid formation of graphite. Only the preferential formation of voluminous graphite achieves a critical thickness to hinder the catalytic effect sufficiently, diamond can nucleate on top of it [14]. The weak bonded graphite at the diamond film/substrate interface usually leads to a spontaneous separation of the diamond films from the substrates. In addition, the difference of thermal expansion coefficients between diamond and the substrates is considerable (at room temperature, α_diamond_ ≈ 0.8 × 10^−6^ /K, α_iron_ ≈ 11.8 × 10^−6^ /K, data from Thermophysical Properties of Matter, The TPRC Data Series), which tends to induce large stresses that crack or delaminate the diamond films from the substrates.

Ferrous-based materials are very significant and widely used in various industrial applications. It is a challenging, but rewarding goal to realize adherent diamond films on steel substrates. Hence, great efforts have been made for CVD growth and the development of high-quality, large-area, adherent diamond films on steel substrates [15,16].

In this review, we first discuss the graphite formation mechanism on steel substrates when the gas phase is a suitable environment for diamond growth, based on the relevant literature and some of our results. Then special attention is paid to the various methods used to suppress the catalytic graphite effects during CVD diamond films on steel substrates. Various interlayers or multi-interlayers have been extensively explored to prevent the graphitization effect, and simultaneously improve the adhesion between diamond films and steel substrates. Meanwhile, some researchers made efforts to directly deposit diamond films on steel substrates without pre-coating interlayers, such as through very particular deposition techniques, precise controls of deposition process, or the approaches of substrate alloying. In the approaches of substrate alloying, we will present our own findings about the unique role of the Al alloying element for improving CVD diamond films on iron alloying substrates. Our results suggest that the way used to improve hot corrosion resistance is also applicable, such as the establishment of Cr_2_O_3_, A1_2_O_3_ or SiO_2_ layers, which are thermodynamically very stable with respect to the metal and have high melting points, whose transport processes through the scales are generally slow. And actually, in the various methods for CVD diamond films on steel substrates, the Al, Si, Cr elements have already been used by some researchers. Recently, amorphous diamond-like carbon (DLC) films on steel substrates have attracted much attention, which can be prepared at very low temperature, and then the precipitated graphite from the internal carburization can be effectively avoided. In addition, based on some of our results, recent advances in the understanding of the mechanism of diamond nucleation are discussed simply.

## 2. Graphite Is the Product of Internal Carburization

During diamond deposition on reactive transition-metal substrates, such as Fe, Co, Ni, a thick graphite interlayer would quickly form on the surfaces of these metal substrates in a very short period of time due to the strong graphite catalytic ability by the Fe, Co, Ni elements. The weak adhesion of the graphite interlayer would cause the following deposited diamond films peeling off from the metal substrates. Mallika et al. [17] and Narayan et al. [18] proposed that the persistence of the graphite signature from Fe, Co, Ni surfaces was related to the lower number of 3d electrons in these metals, which make their reactivity to carbon and the catalytic effects on graphite formation.

In our study, voluminous graphite always formed during diamond deposition on iron and low-alloys such as Fe-Cr, Co-Al substrates. Through the microstructures investigation by cross-sectional transmission electron microscopy (TEM), a severe internal carburization of substrates was observed [19,20], due to the high diffusion coefficient of carbon in these substrates. Then the catalytic formation of graphite by iron, cobalt elements during CVD diamond was clarified. Figure 1 shows an example of severe internal carburization about the iron substrate which was exposed to a CH_4_/H_2_ mixture at about 670 °C during CVD diamond films. The scanning electron microscopy (SEM) image in Figure 1a shows the morphology of voluminous graphite preferentially formed before diamond deposition. Microstructures of the substrates-graphite interfacial region being analyzed by TEM are shown in Figure 1b–d, revealing that on the iron substrate surface, a metastable Fe_3_C carbide layer formed and then decomposed, finally resulting in the precipitation of graphite [20]. The decomposition of a metastable Fe_3_C carbide layer is an important contribution to graphite formation, which is analogous with the well-known effect of metal dusting on iron base substrates [21,22,23].

Based on the results, the graphite formation mechanism is proposed as the follows. During diamond deposition process, a great number of carbon atoms diffuse into the steel substrate. Carbon atoms easily enter to steel substrate because of its high solubility and diffusivity in steel caused also by their inherently small molecules. The severe internal carburization leads to a Fe_3_C carbide layer formed on the substrate surface. The Fe_3_C generation reduces the carbon diffusion from the surface and therefore some graphitic seeds can be built. This triggers the decomposition of the metastable Fe_3_C underneath, because carbon from Fe_3_C wants to go to the graphitic phase. The graphite was precipitated and grew perpendicularly to the surface Fe_3_C carbide, as the carbon atoms from the Fe_3_C decomposition would attach to the basal planes of graphite. Precipitating from the metastable Fe_3_C carbide is the only way for the graphite to form, because the gas phase is favorable for diamond phase formation and the diamond film would grow on the precipitated graphite subsequently.

## 3. The Effective Barriers for the Internal Carburization

Various methods have been employed to suppress the catalytic graphite effects during CVD diamond films on steel substrates, which were essential to hinder the severe internal carburization. These methods mainly contain pre-coating various interlayers or multi-interlayers, precise controls of the deposition process, very particular deposition technique, or the approaches of substrate alloying and so on. 

### 3.1. The Pre-Coating Interlayers on Steel Substrates

A titanium interlayer (about 5–7 μm thick) was sputtered onto the steel wires by Partridge et al., and then a thick (40 μm) and adherent diamond film was successfully deposited by hot-filament CVD [24]. Hoffman et al. found that an intermediate layer consisting of 20 μm thick nitrided chromium on steel substrate was useful. The initial stage of diamond deposition resulted in the partial carburization of the Cr-N interlayer, onto which a continuous diamond film was deposited [25]. Through their further and continuous work, they proposed that the Cr-N interlayer is very promising for industrial application, because this diffusion barrier could not only prevent the catalytic activities of cobalt/iron but also relax the interfacial residual stresses to some extent to enhance the adhesion of diamond films on these substrates [26].

A multilayer of Ni/Cu/Ti was covered on steel substrate by Silva et al. for diamond deposition [27]. Each layer has a specific function: Ni could increase the Cu adhesion to steel substrate, a thick Cu layer accommodated the shear stresses induced by the films/substrate thermal expansion coefficient mismatch, while Ti promoted the CVD diamond nucleation and chemical bonding. A multi-structured Mo-W interlayer was experimented with by Kundrat et al., which could improve the adhesion between diamond films and steel substrates by acting as an effective diffusion barrier during the CVD diamond deposition [28,29].

Although some engineered interfacial layers can greatly suppress the formation of graphite, the majority of them provide unsatisfactory adhesion to both the steel substrates and diamond films. Damm and Contin et al. suggested that the interlayer interposed by laser cladding technique is a promising method for mitigating the stress caused by the large mismatch in thermal expansion coefficient [30]. They used a vanadium carbide thermodiffused coating as the intermediate layer between diamond and steel substrate. This single layer is not only an effective diffusion barrier for carbon and iron, but also has an intermediate thermal expansion coefficient and could be grown to a thickness large enough to mitigate thermal stress [31]. As shown in the Raman spectra of Figure 2, the residual stresses in diamond films are decreased along with the increments of the vanadium carbide intermediate layer.

SiC/Ti and SiC/Cu powder mixtures were also experimented with by Damm and Contin et al. to create the intermediate barrier by a laser cladding technique for diamond film-coated steel [32]. They show good metallurgical bonding with steel and meanwhile, FeSi inner-layer, formed by SiC dissociation, as the main reason for diffusion barrier. A series interlayers such as Cr/CrN, Al/AlSiN and Cr/CrN/CrTiAlN were studied by Li et al. for CVD diamond films on steel substrates [33,34,35]. They found that the multilayer of Cr/CrN/CrTiAlN was efficient to suppress the formation of a crack and its propagation. The aluminum-containing element could enhance diamond nucleation and was beneficial to the deposition of diamond films.

### 3.2. Special Controls of Deposition Process

Some researchers made effort to directly deposit diamond on steel substrates through special controls of deposition process, or using very particular deposition technique.

Zhu et al. reported that diamond deposition on Fe, Co, and Ni has been achieved by a multi-step CVD process consisting of (i) seeding the substrate with diamond powders, (ii) annealing the seeded substrate in hydrogen at high temperatures, and (iii) diamond nucleation and growth [36]. Through the optimization and precise controls of the residence time of source gas during hot-filament CVD, diamond was successfully directly deposited on steel by Nakamura et al. [37] and Kohmura et al. [38], respectively. They suggested that the diamond film with excellent crystalline quality could be obtained by appropriate balance between the growth rate of diamond and the graphite etched rate by hydrogen. The SEM micrographs of diamond grains grown on silicon, stainless steel, and iron substrates under the special controls of the hot-filament CVD process by Nakamura et al. [37] are shown in Figure 3. 

Gowri et al. proposed that diamond films could be successfully deposited on steel substrates without a pre-coating interlayer by the following procedure through using hot-filament CVD [39]. In the first step, a high substrate temperature and a high methane percentage were used to achieve a faster critical carbon concentration and hence a shorter incubation time for diamond nucleation. Subsequently, the substrates were taken out of the reactor and subjected to ultrasonic scratching in diamond slurry in order to increase the diamond nucleation density. Then the final deposition was performed in the reactor under typical diamond growth conditions.

A low-temperature growth technique, 400 °C or lower, in microwave plasma CVD using a surface-wave plasma was reported by Tsugawa et al. [40]. Through this particular deposition technique, nanocrystalline diamond films were successfully deposited on stainless steel substrates without any substrate pretreatments or pre-coating interlayer. However, decreased CVD diamond growth temperature could help to directly deposit diamond films on steel substrates, but the growth rate decreases and the diamond quality is lower, which is not worthwhile.

A new tubular hot-wire CVD system using a tubular quartz vacuum chamber has been designed by Motahari et al. [41]. In their system, the filaments can heat the substrate and act as a gas activator and thermal activator for gas species at the same time. The micro and nanocrystalline diamond films were directly grown on the surface of steel substrates, but the disordered diamond and some non-diamond phases, such as graphitic carbon phases, were also present in the coating films. 

Inzoli et al. found that through a direct-current micro-plasma device, nanocrystalline diamond could be produced on various substrates, even on ferrous materials [42,43]. Using their device, without substrate pre-treatment or interlayer deposition, a continuous nanocrystalline diamond with ballas-like morphology was found on iron substrate after 1 h deposition, and no delamination occurred after substrate cooling. They proposed the obtained adherent diamond film could be attributed to the poor homogeneity of the deposition: in such a case, stress can relax at voids and grain boundaries, preventing delamination from the substrate. 

Narayan et al. found that diamond and a new phase of Q-carbon, harder than diamond, could form by direct conversion of amorphous carbon layers by nanosecond pulsed laser annealing [44]. On steel substrates, they firstly prepared the adherent diamond and Q-carbon layers via nanosecond pulsed laser annealing, and then used these Q-carbon, diamond and Q-carbon-diamond composites as seed layers for hot-filament CVD to grow thicker and more adherent layers of diamond films. The final deposited diamond films on steel substrates are shown in Figure 4. Through their method, the problem that the diamond films suffered from poor adhesion on steel substrates due to the formation of initial soft graphitic layers was solved. In addition, Henda et al. reported that nanocrystalline diamond films could deposit by pulsed electron beam ablation at room temperature under argon background gas, even on steel substrates [45]. Hardness measurements showed film hardness, ranging between 18.5 and 19.5 GPa, which is very similar to the diamond and the new phase of Q-carbon obtained by nanosecond pulsed laser annealing in Narayan et al.’ finding [36].

### 3.3. The Approaches of Substrate Alloying

The substrate alloying was also proposed as a feasible way for diamond films directly depositing on steel substrates without any pre-coating interlayer.

Diamond films deposited on Ni and Ni-Al alloys using hot-filament CVD have been investigated by Narayan et al. [18,46]. They found that the chemical nature of the substrate plays an important role in the stabilization of graphite (sp^2^-bonded) or diamond (sp^3^-bonded) phase. The transition metal substrate of nickel tends to catalyze the formation of graphite. By alloying the substrate with aluminum, it was possible to deposit diamond directly on the NiAl substrates. 

In our study, we find that elements Al and Si in some binary compounds have been shown to be beneficial in reducing the catalytic reactivity of base metals for graphite formation and resulted in enhanced diamond adhesion [47,48,49,50], and in particular the Al alloying of the bulk steel substrates is most effective. In the same microwave plasma-enhanced CVD condition, the behavior of carbon on the Fe-Cr-Al steel substrate and on the Fe-Cr-Ni steel substrate was very different. On the Fe-Cr-Al steel substrate, continuous and adherent diamond film could be directly synthesized, and aligned conical diamond structures were also achieved on this substrate by negatively biasing the substrate holder and inducing a glow discharge. However, on the Fe-Cr-Ni steel substrate, a graphite-rich carbon film incorporated with diamond particles grew in the absence of biasing, then aligned carbon nanotube bundles were formed in the presence of negative biasing and glow discharge [51]. The different deposition behavior of carbon on the two kinds of steel substrates was addressed in terms of the effect of their chemical compositions.

The continuous and dense diamond film directly deposited on the Fe-Cr-Al steel substrate by microwave plasma-enhanced CVD is shown in Figure 5. An enlarged view in the inset reveals that the film is closely packed and has well-faceted crystalline structure with submicron-scaled grain size. A maximum thickness of 3 μm is measured from the cross section (b). The Raman spectrum (c) of this film displays an obvious line splitting at 1344 and 1351 cm^−1^, respectively. The significant upward-shift of the diamond peak from the standard 1332 cm^−1^ is related to high compressive stress in the film. Even so, the adhesion of the diamond film to the substrate appears to be strong enough to withstand the huge load. Surface microhardness of this film-coated Fe-Cr-Al substrate has increased more than 10 times (2883 kg/mm^2^) over the value for the uncoated substrate (260 kg/mm^2^) with a load of 4.9 N. Figure 5d shows an indentation crater under a load of 9.8 N. No spalling or cracking around the film is observed, implying a strong adhesion ability of the diamond film to the substrate.

In order to clarify the fundamental mechanism of the Al alloying element in promoting diamond growth and enhancing interfacial adhesion, fine structure and chemical analysis around the diamond film–substrate interface have been comprehensively characterized by TEM [52]. A comparative study of diamond films formed on different alloy substrates such as pure Fe, Fe-Cr, Fe-Al, and Fe-Cr-Al alloy is conducted. A very thin Al-rich amorphous oxide sublayer is always identified between the diamond film and Fe-Cr-Al substrate interface, as shown in Figure 6. Figure 6a is the cross-sectional TEM image of the adherent film formed on the Fe-Cr-Al substrate, with the electron diffraction pattern (Figure 6b) indicating that the phase in the film is diamond. Figure 6c is an energy dispersive spectrometer (EDS) line scan profile, scanned along from the substrate to the diamond film. The interfacial layer is separated by the vertical lines in the EDS line-scan profile. In this region, the content of element Al is much higher than that in the substrate, whereas there is nearly no Fe and Cr. In the high-resolution TEM (HRTEM) image of Figure 6d, a narrow interfacial layer located between the substrate and the diamond film is clearly seen, which is amorphous and is about 10–15 nm thick. These data imply that a thin Al-rich amorphous interfacial layer with several nanometers in thickness in situ formed along the Fe-Cr-Al substrate/diamond film interface. 

Furthermore, the structures of native oxides formed on Fe-Cr, Fe-Al and Fe-Cr-Al alloys, and the corresponding effects on diamond nucleation and growth in plasma enhanced CH_4_-H_2_ atmosphere were investigated [53]. The ambient environment exposure produces a mixed surface oxide layer containing the oxides of both Fe and Cr or Al on Fe-Cr or Fe-Al alloys, respectively. A selective oxidation of Al occurs on Fe-Cr-Al alloy and it protects the other two substrate elements from oxidation. 

The in situ formation of an amorphous, nanometer scale, Al-rich oxide passive layer at the interface between the diamond film and the substrate plays a crucial role for diamond nucleation, growth, and adhesion, as it acts as an efficient barrier for C diffusion and Fe catalyzed graphite formation. Meanwhile, a synergetic effect of Cr alloying element in Fe-Cr-Al substrate is also useful, which can reduce the fraction of Al, decrease substrate thermal-expansion coefficient, and simultaneously have a mechanical interlocking effect due to the formation of interfacial chromium carbides. Accordingly, a mechanism model is proposed (as shown in Figure 7) to account for the different interfacial adhesion of diamond films grown on the various Fe-based substrates [52].

### 3.4. Methods Similar to Hot Corrosion Resistance, Such as Aluminizing, Siliconizing etc.

The above studies, especially the severe internal carburization causing the voluminous graphite precipitated from the steel substrates in Section 2, and the substrate alloying by the Al element being efficient for suppressing the graphite formation during diamond film deposited on Fe-Cr-Al substrate in Section 3.3, suggest that the way used to improve hot corrosion resistance is also applicable for diamond films deposited on steel substrates. Many alloys or alloy coatings for high-temperature applications are based on iron, nickel and/or cobalt and rely on the establishment of Cr_2_O_3_, A1_2_O_3_ or SiO_2_ healing layers for protection against oxidation. These oxides are thermodynamically very stable with respect to the metal and have high melting points; transport processes through the scales are generally slow [54,55,56,57]. Actually, in the various methods for CVD diamond films on steel substrates, the Al, Si, Cr elements have already been used by some researchers.

Siliconization of steels has been studied previously as a way to improve hot corrosion resistance, while Alvarez et al. used the similarly method to prepare silicon diffusion surface interlayers on steels [58] for diamond deposition. By coating the surface of the steels with silicon and performing diffusion treatments at 800 °C, a uniform 2 μm thick silicon rich interlayer was obtained. They found that it was an efficient barrier, and the continuous and adherent diamond films were successfully deposited on the diffused silicon interlayer. Ong et al. found that a thin (~200 Å) silicon buffer interlayer was effective in inhibiting surface catalytic effect of iron and also prevented carbon species from diffusing into the bulk [59]. The reason for this they proposed was as follows: even though the hydrogen etching of the silicon barrier takes place at all substrate temperatures, at high (>600 °C) substrate temperatures the silicon interlayer would start to oxidize due to the presence of the oxygen in the plasma. Once the oxide layer is formed, the hydrogen etch rate of the silicon barrier is dropped by about an order of magnitude. This oxide layer quenches the thickness reduction process of the barrier for nucleation of the diamond on the iron surface. Also, Reinoso et al. reported that through pretreating by means of a silicon ion beam, a thin silicon layer was coated on steel substrates, and then the adherent amorphous carbon (a-C) and diamond films had been deposited on steel substrates [60].

We have undertaken long, continuing and relevant work into the role of Al in CVD diamond films on steel substrates. Based on our surprising findings about the effective role of Al alloying element [47,48,49,52], we tried to solve the problem that most steels did not contain Al. In our work, firstly, a thin Al interlayer was precoated on steel substrate for diamond film deposition. However, using only a simple Al thin interlayer is not sufficient, and the failure mechanism of adherent diamond film at local positions was elucidated [61]. The main reason is that the integrity of the Al interlayer is easily destroyed during scratching pretreatment, consequently, the continuity of diamond film is damaged and local carburization corrosion occurs on the substrate. Then the dual interlayers of Cr-Al, Ti-Al and W-Al were tested for diamond films deposited on steel substrates [62,63]. We found that an ultrathin W-Al dual interlayer is very promising. In addition, hot dip aluminizing treatment of the steel or Al-ion implanted steel were also carried out in our experiments [64,65]. Through these means, the diamond nucleation, growth and interfacial adhesion have been markedly promoted. 

Ye et al. [66] used a Cr_2_O_3_-Cr duplex interlayer, in which Cr_2_O_3_ layer was an effective barrier to hinder the graphite formation, and the top Cr layer was carburized to Cr_3_C_2_ and Cr_7_C_3_ to enhance the interfacial adhesion. It is worthwhile to note that in some jobs about high temperature corrosion resistance, Fe-Cr-Al has been used to coat steel substrates and increase their corrosion resistance [67,68]. This may be also a promising interlayer for CVD diamond films on steel substrates.

### 3.5. Another Way—Prepared Diamond-Like Carbon (DLC) Films 

The common temperatures for preparing crystalline diamond films by CVD are about 700–900 °C. However, in this temperature zone, the internal carburization of steel substrates cannot be avoided due to the high carbon diffusion in iron. The above reviewed methods are to hinder the internal carburization by various barriers. Recently, many researchers have also been focusing on preparing diamond-like carbon (DLC) films on steel substrates, which could be prepared at low temperatures (mostly about 200–400 °C) [69,70,71,72,73], and accordingly, the problem of internal carburization is effectively solved [74].

DLC is an amorphous carbon material with mostly sp^3^ bonding, which exhibits many of the desirable properties of the diamond material. In the work of Ghorbani et al., using a high-energy plasma focus device, DLC films were deposited from the mixture of C_2_H_2_/H_2_ gas at room temperature on steel (AISI 304) substrates [75]. Through the control of the hydrogen proportion in the mixture gas, the effect of hydrogen content on the structural and mechanical properties of DLC films was analyzed by them. In the work of An et al., pulsed kV bias technique was applied to prepare DLC films, which can increase the carbonous ion energy to increase the sp^3^ content in DLC films and improve the adhesion with the substrate simultaneously while avoiding adverse temperature increase [76]. Some other methods, such as preparing DLC films on steel by plasma-enhanced CVD at lower temperature (about 300 °C) [77,78], and preparing DLC films on inner surfaces of steel tubes by nanopulse plasma CVD [79], were also reported. In the work of Yang et al., the catalytic effect of Fe_3_C on DLC growth was investigated [80]. Through a single-step plasma-assisted carburizing process, the Fe_3_C-containing carburized surface layer was formed on steel (M50NiL). They found that with help of the catalytic effect of Fe_3_C, DLC could simultaneously form during carburization of steel under suitable processing conditions, resulting in a combination of DLC and carburized layers, which is a promising approach to maximizing the benefits of carburization treatment, and provides new clues for facilitating DLC production and improving traditional surface treatments for steels.

Meanwhile, in order to improve the interfacial adhesion, some interlayers were also introduced between DLC films and steel substrates. A Cr/Ti/TiC graded interlayer was introduced on steel substrate by Galvan et al., and then a TiC/DLC nanocomposite film was deposited on this graded interlayer via magnetron sputtering in an Ar/C_2_H_2_ atmosphere [81]. The detailed microstructure of the graded interlayer and the film was investigated by cross-sectional TEM. Figure 8 shows the interfacial microstructures, a satisfactory composition gradient is produced, which leads to a good adhesion between the film and the substrate. A silicon-containing interlayer was introduced by Cemin et al. during DLC deposition on steel (AISI 4140) [82], which could enhance the interfacial adhesion. Interestingly, after growing of the silicon-containing interlayers, the DLC films could be deposited at very low temperature, even possibly at the substrate temperature that reached 80 °C in a gaseous mixture of Ar/C_2_H_2_. Assala et al. [83] introduced a nitriding layer by plasma nitriding of steel substrates (41Cr-Al-Mo7), and then DLC films were grown on nitrided steel by microwave electron cyclotron resonance/plasma-assisted CVD with a C_6_H_6_-Ar gas mixture. An observed increase in adhesion was obtained between DLC films and nitrided steel substrates, which could be attributed to the final formation of the interlayer phases such as (Fe,Cr)_3_C, Cr_3_C_2_ and Cr_23_C_6_.

Another interesting topic is doped DLC films on steel substrates. Doped DLC films have been prepared and studied, such as B-doped [84,85], Ti-doped [81], Ag-doped [86,87], Ge-doped [88] and so on. These special doped DLC films are ideal protection for the biomedical alloys from oxidation, passivation and to reduce the ability for a bacterial biofilm to form after implantation [89]. Ag-containing DLC films were prepared on 316 steel substrates by magnetron sputtering at different silver target currents, which showed a long-lasting and reusable antibacterial activity through the antibacterial tests of Escherichia coli, attributed to the Ag particles distributed in the films [86,87]. Lopes et al. introduced titanium dioxide nanoparticles into the DLC films grown on 304 steel by CVD [90]. They found that the antibacterial properties of the DLC films were significantly improved.

## 4. Diamond Nucleation on Steel

It is an interesting and yet intriguing problem why meta-stable diamond can be grown on diamond or non-diamond substrates under CVD conditions. To resolve the problem, the substrate surface must be characterized thoroughly since nucleation is a surface phenomenon. It is well known that the most stable carbon phase is graphite and that diamond is metastable. Although the energy difference between the two phases is only 0.01 to 0.04 eV/atom, the high activation barrier (~0.4 eV/atom) is required to achieve the phase transition, which needs extreme conditions of temperature and pressure.

According to CVD diamond literature, diamond can homoepitaxially nucleate and grow on diamond substrate [91], and meanwhile heterepitaxially nucleate and grow on graphite, Si and other substrates [92,93,94]. However, the nucleation on various non-diamond substrates is much different. A model for diamond nucleation by energetic species (for example, bias enhanced nucleation) was proposed by Lifshitz et al. [95]. It involves spontaneous bulk nucleation of a diamond embryo cluster in a dense, amorphous carbon hydrogenated matrix; stabilization of the cluster by favorable boundary conditions of nucleation sites and hydrogen termination; and ion bombardment-induced growth through a preferential displacement mechanism. 

Brescia et al. [96] detected the diamond nuclei generated during bias enhanced nucleation on iridium by HRTEM. The earliest appearance after bias enhanced nucleation by applying very short growth steps, ranging from 5 s to 1 min, was investigated. During bias-enhanced nucleation a 1–2 nm thick amorphous carbon layer was formed on the iridium, and diamond nucleation took place at few spots in the amorphous carbon layer. While, after termination of the bias, the amorphous carbon layer became unstable and they were completely etched within few seconds (5–10 s). 

Wang et al. used [97] ultrananocrystalline diamond (~10 nm in size) as nucleation layer for growing diamond films. They also found that a thin layer of amorphous carbons was formed first, and then the ultrananocrystalline diamond grains nucleated heterogeneously at the amorphous-to-Si interface rather than homogeneously inside the amorphous layer. Such a phenomenon is understandable, as the heterogeneous nucleation process is always energetically favorable, as compared to the homogeneous nucleation one.

In our work [98], through the detailed cross-sectional TEM investigation of the interfacial region between diamond and steel substrates, the results also support the nucleation for the energetic amorphous carbon, as the mechanism proposed by Lifshitz et al. [95].

The interfacial region from iron substrate, graphitic interlayer to diamond film is shown in Figure 9. The cross-sectional TEM images of the delaminated diamond film and the residual graphitic interlayer on iron substrate are shown in Figure 9a,b, respectively. An obvious difference was observed between the two sides of the delaminated diamond film, from the poor and loose graphite side to the dense and continuous diamond film side. Many unique diamond clusters with a diameter about 400–500 nm locate in the transitional zone, which act the initial diamond nucleation roles. One of these clusters is shown in Figure 9c, with its corresponding selected-area electron diffraction (SAED) pattern shown in Figure 9d. The diffraction rings in the SAED pattern reveal that many fine diamond grains are contained in the clusters. Although the initial nucleation sites for diamond must be contained in these diamond clusters, it is difficult to clearly identify the initial nucleation sites for diamond. That is because through the HRTEM study of the cluster (as shown in Figure 9e), the structure of the cluster is very complex containing many diamond grains, graphite and carbide particles.

The interfacial region between initial diamond and Fe-Cr-Al substrate is shown in Figure 10. As an amorphous thin Al-rich oxide layer would in-situ quickly form on Fe-Cr-Al substrate, the formation of a graphite interlayer is effectively hindered. The Fe-Cr-Al substrate maintains as α-Fe phase, and no Fe_3_C carbide forms on the substrate surface, by contrast with that on iron substrate which forms a thick Fe_3_C carbide layer. Consequently, the precipitation of graphite from Fe_3_C carbide cannot occur on Fe-Cr-Al substrate. Instead, diamond could directly nucleate and grow on the Fe-Cr-Al substrate surface. The SEM image in Figure 10a is the sample being deposited for very short time, showing that at the very early stage only some small diamond particles dispersed on the Fe-Cr-Al substrate surface. Luckily, some unique phenomena of this sample are observed in the cross-sectional TEM investigation, as shown in Figure 10b–d. In the surficial groove (Figure 10b), there are a large number of elongated particles. Its SAED pattern in Figure 10c shows that the diffraction rings are corresponding to (111), (220) and (311) of diamond respectively, revealing that many small diamond grains are formed in this region. The elongated particles are distributed randomly, 10–15 nm in length and about 5 nm in width. One of the elongated particles is presented in the HRTEM image of Figure 10d, with the inset showing the corresponding FFT. Both the lattice spacing and angle are consistent with the (111) plane of diamond in the HRTEM image of the elongated particle. Meanwhile, the corresponding diffraction pattern in FFT can be indexed as [110] zone axis of diamond. These fully demonstrate that the elongated particles are diamond phase. It should be noted that the elongated diamond particles cannot be the pre-existing diamond seeds, as scratching by diamond paste were not carried out on the substrate surface prior to diamond film deposition. These particles are surrounded by an amorphous carbon matrix, suggesting that the diamond phase nucleates from the amorphous carbon.

In our group’s previous work [99], we found that diamond films (on silicon wafers) and carbon nanotubes (on Inconel plates) could be simultaneously synthesized without any additional catalyst through a microwave plasma-enhanced CVD reactor in a CH_4_/H_2_ gas mixture. This means that the CVD gas phase is favorable for diamond formation, while the graphite formation is due to the substrate catalytic effect. Once the catalytic graphite effect by the substrate is effectively hindered, just like our above Fe-Cr-Al substrate, only diamond films deposited. During the CVD initial stage, the amorphous Al-rich oxide layer quickly formed in situ can effectively hinder the carbon atoms diffusing into the substrate, and consequently, large number of carbon will accumulate at the surface, especially, a quick accumulation of carbon occurs at the surface groove. With the penetration of hydrogen atoms from gas, a dense amorphous hydrogenated carbon (C:H) phase form in the groove, then, diamond nanocrystallines would precipitate from these amorphous carbon, which is consistent with the spontaneous nucleation proposed by Lifshitz et al. [95].

During the study of Sun et al. [100] on the reductive alkylation of anthracite by dodecyl groups, they found that diamonds were formed when the dodecylated anthracite was exposed to the electron beam during analysis by TEM. Figure 11 shows the small diamonds formed from anthracite under electron irradiation. Although their comparatively study with pristine anthracite failed to yield diamonds, the diamond formation mechanism is clarified. They proposed that the hydrogen plays a dominant role, which is similar to CVD growth, a dense amorphous hydrogenated carbon (C:H) phase forms during hydrogen atoms penetrate the graphitic-like layers of anthracite. After the carbon was saturated by hydrogen, precipitation of sp^3^ carbon clusters would occur. Most of the clusters would probably be amorphous, but a few would have the structure of diamond. The formation of diamond observed during their study is consistent with the model proposed and substantiated by Lifshitz and coworkers [95]. Also, since our synthetic diamond films were achieved in a CVD reactor with a CH_4_/H_2_ gas mixture, our observed [98] diamond nuclei on Fe-Cr-Al substrate is also consistent with their proposed mechanism that diamond phase nucleates from the energetic amorphous carbon.

## 5. Summary

The synthesis of diamond films by chemical vapor deposition (CVD) has served as a great breakthrough and made many applications of diamond possible. In particular, CVD diamond films on steel substrates have attracted great attention. Since the steels are the most commonly used and cost effective structural materials in modern industry. CVD coating diamond films on steel substrates is extremely important, which will combine the unique surface properties of diamond with the superior toughness and strength of the core steel substrates, and will open up many new applications in industry. However, voluminous catalytic graphite formation on steel substrates before diamond deposition is the main drawbacks and CVD adherent diamond films on steel substrates continues to be a persistent problem. 

The main reason for the catalytic graphite formation is analyzed. The high carbon diffusion from gas phase into steel substrates causes severe substrate carburization. A metastable Fe_3_C carbide layer forms on the steel substrate surface due to the severe internal carburization. Then the carbide would further decompose, which is an important contribution to graphite precipitation. 

Naturally, to solve the problem of catalytic graphite and meanwhile guarantee the diamond film’s adherence is a big challenge, which has attracted many researchers to this study. In order to solve such complicated and difficult problems, various techniques have been applied to hinder the internal carburization. Many efficient interlayers or multi-interlayers as a diffusion barrier for both, carbon and iron have been proposed in the literature. Meanwhile, special controls of the CVD process have been developed for the direct deposition diamond on steel substrates. Through the approaches of substrate alloying, adherent diamond films have been directly deposited on steels by our group. Our further analysis illustrates that the Al alloying element plays a critical role by in situ forming a thin amorphous alumina sublayer, suggesting that the way used to improve hot corrosion resistance is also applicable for diamond films depositing on steel substrates. Actually, in literature, some researchers have already used the Al, Si, Cr elements to establish A1_2_O_3_, SiO_2_ or Cr_2_O_3_ healing layers to protect the steel substrate surface during CVD diamond film creation. It is worth noting that amorphous diamond-like carbon (DLC) films prepared on steel substrates at very low temperature have attracted many attentions recently. According, the above problem of internal carburization can be effectively solved. While how to control the DLC films to exhibit the desirable properties of diamond is still a challenge.

Finally, the diamond nucleation is studied. The diamond nucleation from energetic amorphous carbon is an important method, and direct evidence has been provided by some researchers. The atomic hydrogen and the concentration of hydrocarbon radicals play a critical role in the formation of diamond nuclei. 

## Figures and Tables

**Figure 1 micromachines-11-00719-f001:**
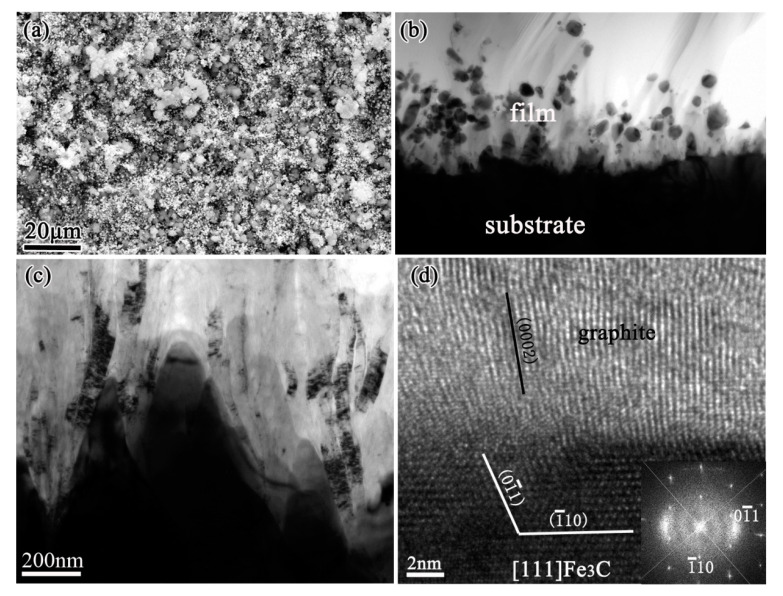
A thick graphite layer formed on iron substrate before diamond deposition: (**a**) scanning electron microscope (SEM) image, (**b**,**c**) cross-sectional transmission electron microscope (TEM) images of the interface between iron substrate and graphite layer, showing the graphite precipitated at the surfacial grooves of iron substrate. (**d**) High-resolution TEM (HRTEM) image from these grooves, with the corresponding Fast Fourier Transform (FFT) inserted. (Reproduced with permission from [20]).

**Figure 2 micromachines-11-00719-f002:**
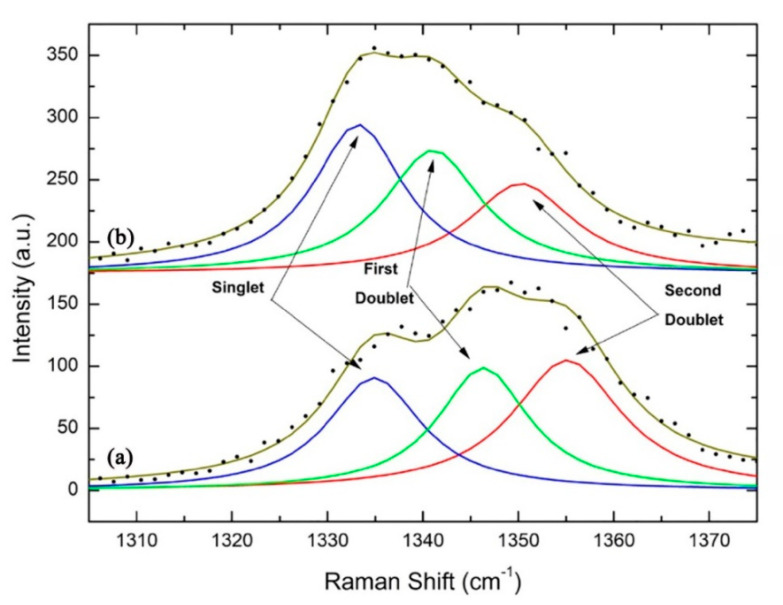
The singlet and doublets behavior on diamond film residual stress: (**a**) with 9 μm and (**b**) with 31 μm of the vanadium carbide intermediate layer. (Reproduced with permission from [31]).

**Figure 3 micromachines-11-00719-f003:**
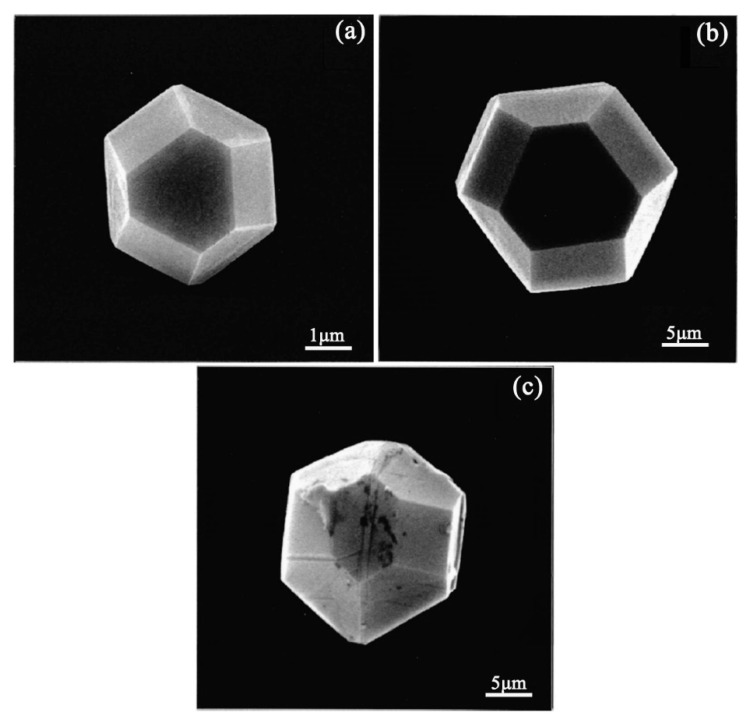
SEM images of diamonds grown on: (**a**) Si substrate, (**b**) stainless steel substrate, and (**c**) iron substrate at a residence time of 6 ms. Growth time is 3 h for all substrates. (Reproduced with permission from [37]).

**Figure 4 micromachines-11-00719-f004:**
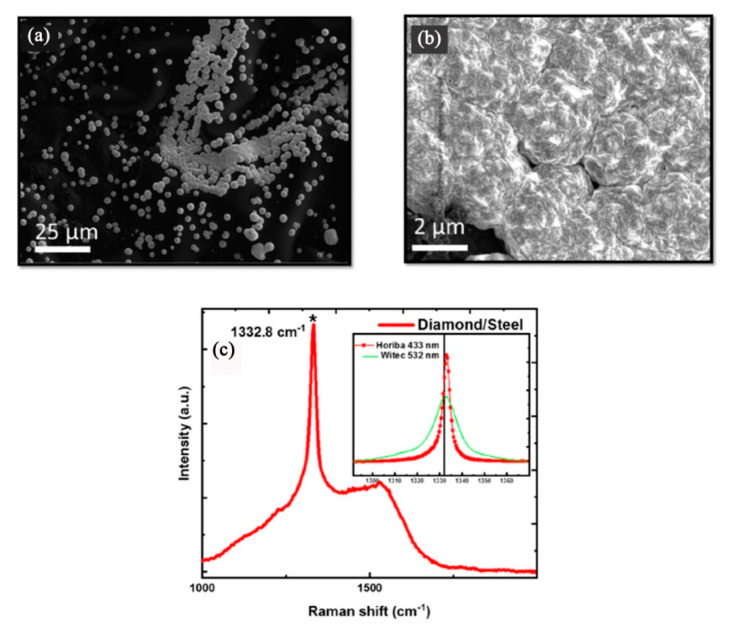
Microdiamonds can formed on steel substrates by hot-filament chemical vapor deposition (CVD), using the Q-carbon seeds formed after single-pulse pulsed laser annealing at 1.1 J/cm^2^: (**a**) SEM micrograph (**b**) The enlarged SEM micrograph and (**c**) Raman spectrum of the formed microdiamonds. (Reproduced with permission from [44]).

**Figure 5 micromachines-11-00719-f005:**
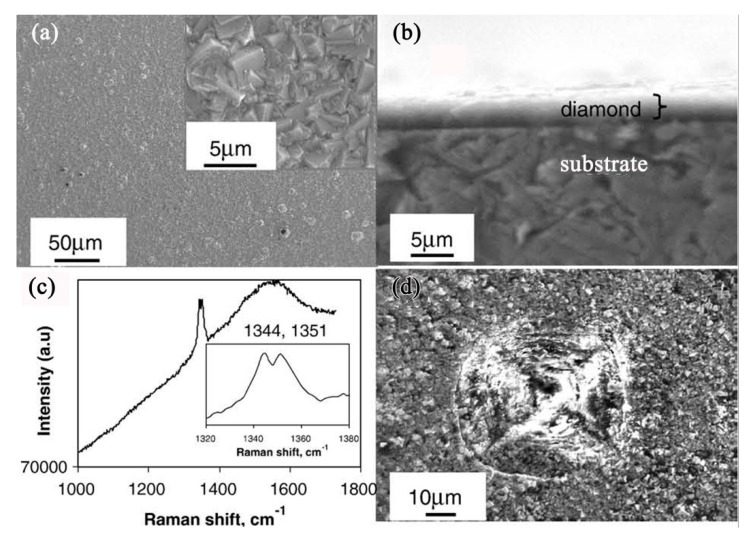
SEM images of diamond film deposited for 12 h on Fe-Cr-Al substrate: (**a**) general and magnified (inset) views of the diamond film; (**b**) a cross sectional view; (**c**) Raman spectrum showing the exact peak position (inset); (**d**) a top view of the film after indentation test at a load of 9.8 N. (Reproduced with permission from [48]).

**Figure 6 micromachines-11-00719-f006:**
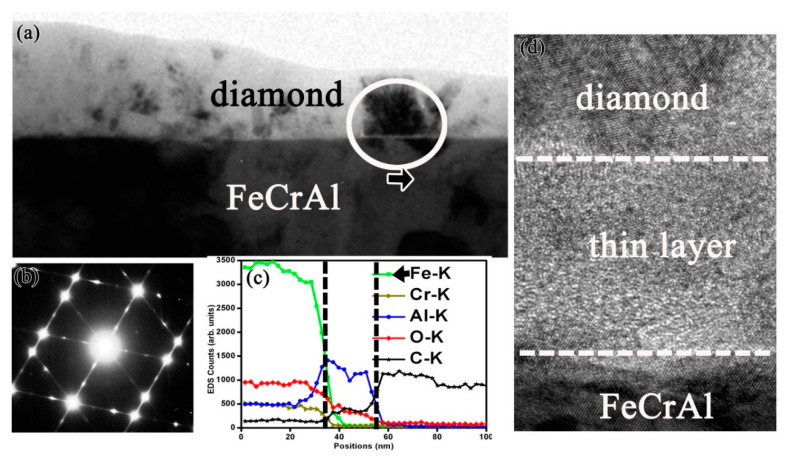
(**a**) Cross-sectional TEM image of the diamond film prepared on the Fe-Cr-Al substrate; (**b**) Diffraction pattern of the diamond film; (**c**) EDS line-scan profile scanned along from the substrate to the film. There exists an interfacial layer where Al element content is much higher. (**d**) The interfacial HRTEM image, the arrow indicates the location where the diamond crystalline starts growing on top of the amorphous phase. (Reproduced with permission from [52]).

**Figure 7 micromachines-11-00719-f007:**
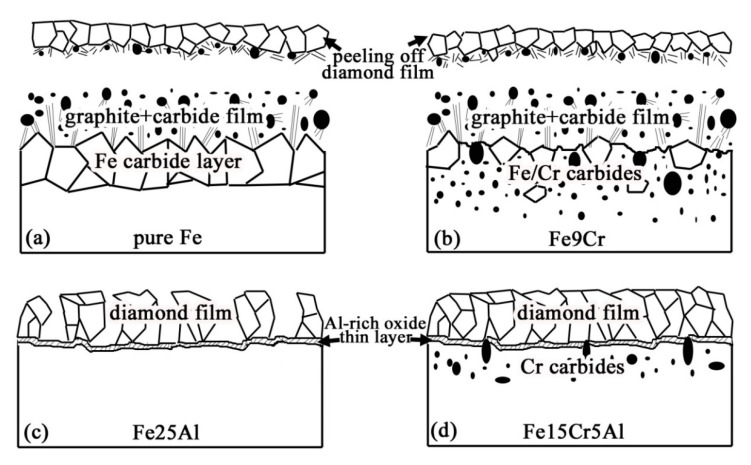
Model describing the mechanism of diamond films grown on Al, Cr-modified Fe-based substrate is proposed. (Reproduced with permission from [52]).

**Figure 8 micromachines-11-00719-f008:**
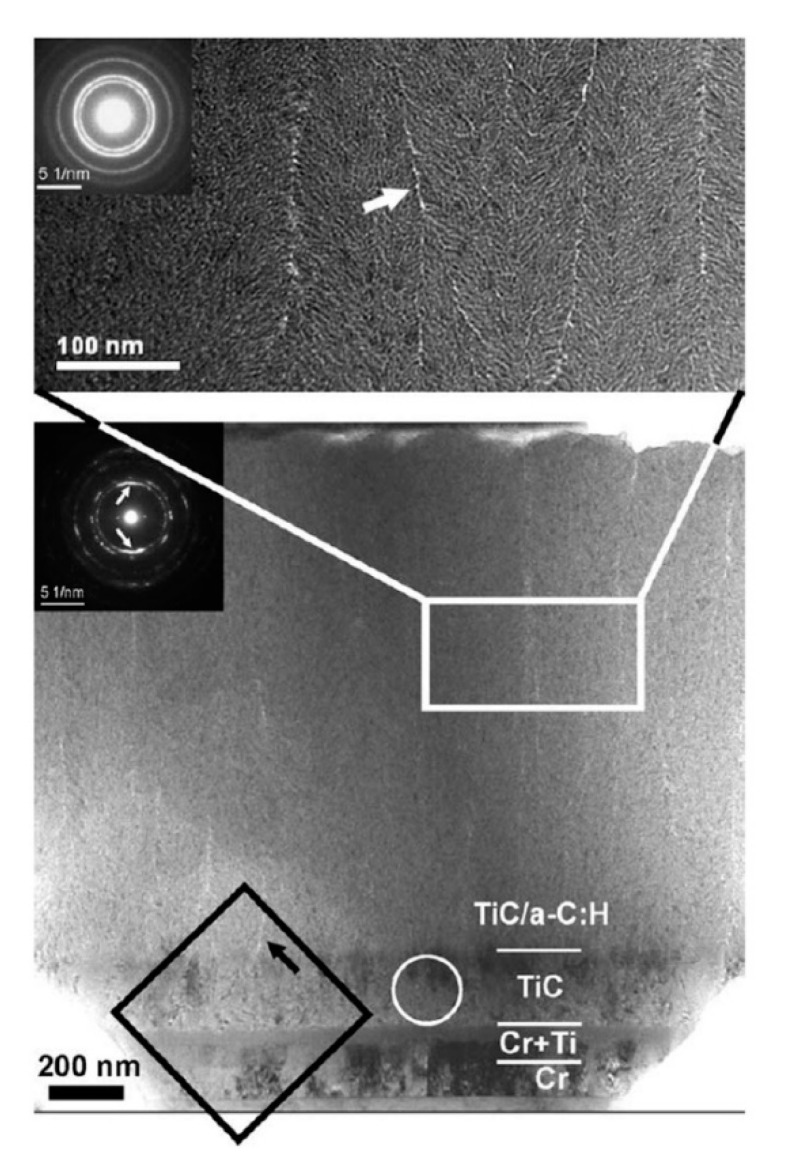
Cross-sectional TEM image of the TiC/diamond-like carbon (DLC) nanocomposite film deposited on steel substrate through a Cr/Ti/TiC graded interlayer. (Reproduced with permission from [81]).

**Figure 9 micromachines-11-00719-f009:**
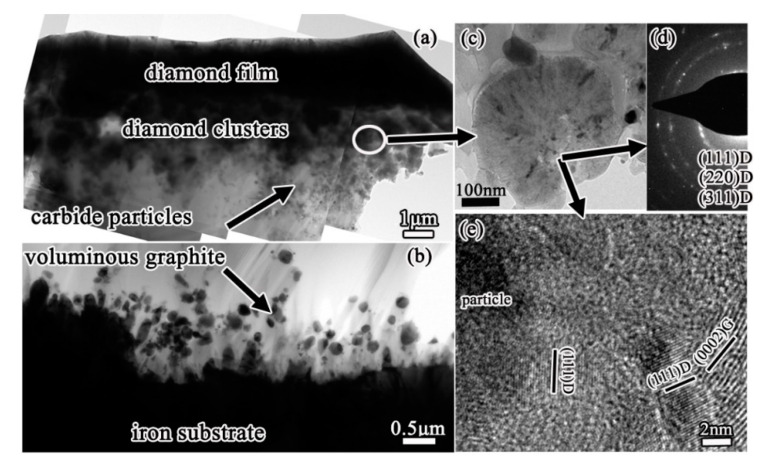
Cross-sectional TEM images of (**a**) the delaminated diamond film peeled off and (**b**) the residual graphite interlayer on iron substrate. Some clusters are observed in the transitional zone: (**c**) enlarged-TEM image, (**d**) corresponding selected-area electron diffraction (SAED) pattern, and (**e**) HRTEM image of the cluster. (Reproduced with permission from [98]).

**Figure 10 micromachines-11-00719-f010:**
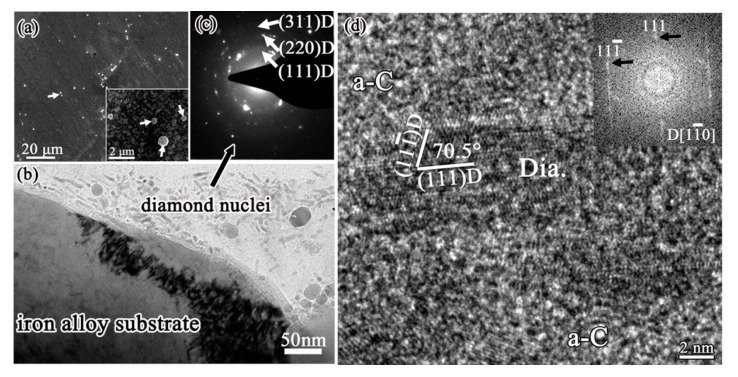
Diamond deposited on Fe-Cr-Al substrate at the very early stage: (**a**) SEM image, with the inset showing the magnified view, showing only some small diamond particles disperse on the substrate surface; (**b**) cross-sectional TEM image shows one surfacial grooves on Fe-Cr-Al substrate, with some diamond nuclei were found; (**c**) the corresponding SAED pattern of these diamond nuclei; (**d**) HRTEM image of one of these diamond nuclei, with the inset showing the corresponding FFT. (Reproduced with permission from [90]).

**Figure 11 micromachines-11-00719-f011:**
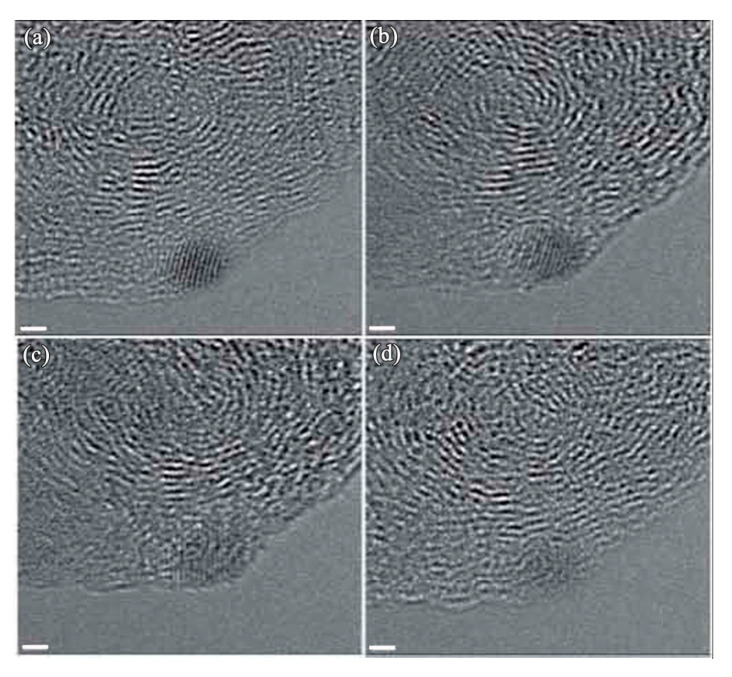
HRTEM images showing a nanodiamond forms from the anthracite upon exposure to the electron beam. The scale bar is 1 nm (Reproduced with permission from [100]).

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
