# Peer review of "Diamond Deposition on Iron and Steel Substrates: A Review"

_micromachines, 2020, doi:10.3390/mi11080719_

Round 1

Reviewer 1 Report

Abstract should be shortened. It should be more concise. For example, you should formulate the problem and how to solve it. Do not get into diamond nucleation mechanisms on different interlayer. This should be moved to the main text.

The organization of the main text is very chotic. The authors discuss various types of transition layers on a steel substrate, e.g. aluminizing, siliconizing, catalytic graphite interlayer, DLC and so on.

I suggest that each of these methods be discussed separately. In addition, the same hardness measurement units must be used.

Additionally, the text contains many repetitions.

Line 62 nad 96 - surfaces of these metal substrates would turn black

Line 107- which was exposed to a CH4-H2 mixture at about 670 C during CVD diamond films – the sentence is incomplete

Line 151- what is precipitating from the metastable Fe3C carbide ???- rewrite

Line 148-154: it is not clear whether the formation of the graphite phase is a beneficial process or not. This must be expressed more clearly

Fig.2. - You have to comment on this more. If you see the splitting of the Raman line i.e. there is a high internal stress. How big is it?

Line 208- shoud be rewritten

Line 209- good metallurgical bonding?? – such term does not exists. Probably adhesion??

Line 212- for CVD diamond films on steel substrates?? - for CVD diamond films growth on steel substrates

Line 213- it's probably not about high toughness and high hardness but about the coefficient of thermal expansion

Line 250- etching rate of hydrogen??, Hydrogen is not eatched!!!,  hydrogen is eatching something

Line 264-266 – rewrite

Line 271 - graphitic carbons??, graphitic carbon phase

Fig.4 - explain why FWHM’s  of Raman lines are different at different excitations

Line 326 - The substrate alloying was also proposed be a way…-  The substrate alloying was also proposed to be a way….  

Line 330-331-  grammar

Line357 – grammar

Line 377- Fig.d????

Line 410 – Fig.4a???

Lie 420-422- what electronic structure is it about?

Line 453- interlocking????

Line 477- the hydrogen etch rate  of what???

Line 482 – not precise! is it Al as an element? do you need to supplement something?

Line 487-489  - it is very obvious

Line 516-517 – repetition of „to prepare” – rewrite

Line 528-529- rewrite

Line 563 - requires deeper commenting

Line 583 -586 - hetroepitaxy is a synthesis on substrates with a similar lattice constants!!!!.

Author Response

We thank a lot for the valuable comments. The manuscript has been revised based on the suggestions. The following are our point-to-point responses to all the comments, and responses are in blue.

Responses to Reviewer 1

  1. Abstract should be shortened. It should be more concise. For example, you should formulate the problem and how to solve it. Do not get into diamond nucleation mechanisms on different interlayer. This should be moved to the main text.

Response: It may be not proper to not state the diamond nucleation in the abstract. As a relatively part in the review paper discusses the problem of diamond nucleation.

  1. The organization of the main text is very chotic. The authors discuss various types of transition layers on a steel substrate, e.g. aluminizing, siliconizing, catalytic graphite interlayer, DLC and so on. I suggest that each of these methods be discussed separately. In addition, the same hardness measurement units must be used.

Response: We try our best to discuss the methods separately. But some methods such as aluminizing, siliconizing and so on are essential the same, as they are rely on the establishment of A12O3 SiO2 or Cr2O3 healing layers for protection, being the similar way to improve hot corrosion resistance.

It is a big challenge for us to make the same hardness measurement units. That is because it is usually different in the different researcher jobs.

  1. Additionally, the text contains many repetitions.

Line 62 and 96 - surfaces of these metal substrates would turn black

Response: The repetition in Line 96 has been amended in the revised paper.

  1. 4. Line 107- which was exposed to a CH4-H2 mixture at about 670 C during CVD diamond films – the sentence is incomplete

Response: This sentence has been amended in the revised paper.

  1. Line 151- what is precipitating from the metastable Fe3C carbide ???- rewrite

Response: This sentence has been rewritten in the revised paper.

  1. Line 148-154: it is not clear whether the formation of the graphite phase is a beneficial process or not. This must be expressed more clearly

Response: It is a bad process for the CVD diamond films on steel substrates. In the revised paper, we have express this problem more clearly.

  1. Fig.2. - You have to comment on this more. If you see the splitting of the Raman line i.e. there is a high internal stress. How big is it?

Response: The Raman shift from 1332 cm-1 can be used to estimate the stress. The shift is larger, the stress is bigger. But how big is very difficult to precise estimation.

  1. Line 208- shoud be rewritten

Response: This sentence has been rewritten in the revised paper.

  1. Line 209- good metallurgical bonding?? – such term does not exists. Probably adhesion??

Response: We propose the “good metallurgical bonding” is proper, as the Si element will diffuse into the steel substrate during laser cladding technique, and then a FeSi inner layer can form, which is according to the metallurgical bonding.

  1. Line 212- for CVD diamond films on steel substrates?? - for CVD diamond films growth on steel substrates

Response: We suggest this expression of “for CVD diamond films growth on steel substrates” is proper.

  1. Line 213- it's probably not about high toughness and high hardness but about the coefficient of thermal expansion

Response: This expression has been amended in the revised paper.

  1. Line 250- etching rate of hydrogen??, Hydrogen is not eatched!!!,  hydrogen is eatching something

Response: This expression has been amended in the revised paper.

  1. Line 264-266 – rewrite

Response: This expression has been rewritten in the revised paper.

  1. Line 271 - graphitic carbons??, graphitic carbon phase

Response: This expression has been amended in the revised paper.

  1. Fig.4 - explain why FWHM’s of Raman lines are different at different excitations

Response: Indeed, it is difficult for us to accurate explain the reason.

  1. Line 326 - The substrate alloying was also proposed be a way…-  The substrate alloying was also proposed to be a way….  

Response: This expression has been amended in the revised paper.

  1. Line 330-331- grammar

Response: The grammar has been amended in the revised paper.

  1. Line357 – grammar

Response: The grammar has been amended in the revised paper.

  1. Line 377- Fig.d????

Response: This expression has been amended in the revised paper.

  1. Line 410 – Fig.4a???

Response: This expression has been amended in the revised paper.

  1. Line 420-422- what electronic structure is it about?

Response: This expression has been amended to “structure” in the revised paper.

  1. Line 453- interlocking????

Response: We suggest this expression is proper.

  1. Line 477- the hydrogen etch rate of what???

Response: This expression has been amended in the revised paper.

  1. Line 482 – not precise! is it Al as an element? do you need to supplement something?

Response: This expression has been amended in the revised paper.

  1. Line 487-489 - it is very obvious

Response: We suggest that the obvious reason is also worthwhile to describe.

  1. Line 516-517 – repetition of „to prepare” – rewrite

Response: This expression has been rewritten in the revised paper.

  1. Line 528-529- rewrite

Response: This expression has been rewritten in the revised paper.

  1. Line 563 - requires deeper commenting

Response: We suggest a deeper understanding of this job can refer the referred job.

  1. Line 583 -586 - hetroepitaxy is a synthesis on substrates with a similar lattice constants!!!!.

Response: But indeed in the referred job, the heterepitaxial nucleation and growth on graphite, Si and et al substrates have been reported by some researchers.

Reviewer 2 Report

I find this review very interesting for the diamond community. However, before I can recommend it for publication a couple of minor corrections have to be done;

1) The font sizes are different in the various Figures. If possible, make them all to be, at least, of the same size.

2) The English language has to be improved.

Author Response

We thank a lot for the valuable comments. The manuscript has been revised based on the suggestions. The following are our point-to-point responses to all the comments, and responses are in blue.

Responses to Reviewer 1

I find this review very interesting for the diamond community. However, before I can recommend it for publication a couple of minor corrections have to be done;

  1. The font sizes are different in the various Figures. If possible, make them all to be, at least, of the same size.

Response: The font sizes in the Figures (Figures 2, 3, 4, 5, 11) have been relabeled. They look like at the same size in the revised paper.

  1. The English language has to be improved.

Response: We have revised the WHOLE manuscript carefully. And we have tried to avoid any grammar or syntax error in the revised paper.

Reviewer 3 Report

The present work aims to present a review on the subject of “CVD diamond deposition on iron and steel”, as stated in the title. This is a first problem: nobody does “diamond deposition on iron”. What is technological relevant is “diamond deposition on steel”. Fundamental studies using metallic iron or iron alloys could be important to understand the phenomena when steel is used as substrate but are secondary. So, the title needs revision.

As authors know, the goal of “CVD diamond deposition on steel” is to obtain: (1) dense, (2) uniform, (3) adherent CVD diamond coatings, with (4) good quality diamond. So, a review on this subject should present and discuss the main strategies to achieve that. And this should be tentatively done in comparative tables rather than by sequential resumes of papers.

I liked the organization of the manuscript: starting by the graphitization problem in section 2, followed by section 3 with different approaches to hinder the phenomenon, appearing as subsections 3.1. to 3.5. However, I found the last one (“3.5….DLC films”) completely outside the subject because DLC has nothing to do with diamond and a review on “DLC on steel substrates” would be another manuscript. This is confirmed if you do a search in Web of Science for TITLE (dlc) AND TITLE (steel) where you find 202 papers. So I suggest to remove subsection 3.5.

Another major fault is the exaggerated emphasis the authors put in their previous work, exactly in what is not so relevant: the mechanism of diamond nucleation & growth on a Fe-Cr-Al ternary alloy. Figures 5, 6, 7, 9 and 10 refer to previous results of the authors and are excessive for a balanced presentation of the subjects. It doesn't make sense, for example, that section “4. Diamond nucleation on steel” is based on a paper about nucleation on Fe-Cr-Al.

Other amendments are suggested in the following:

1- The manuscript requires significant proof reading and revision to improve the quality of English.

2- Section 2: graphite formation mechanism is well known and authors cannot state that they want to propose a new one. Refs [14] and [39] also discuss the question of Fe3C formation as also does: Polini et al, Thin Solid Films 494 (2006) 116-122.

3- Section 2: Figures 1(b) and (c) are very similar and I don’t see a “film” in figure 1(b). Where is the Fe3C layer in Figures 1(b) or 1(c)?

4- Section 3: regarding Figure 2, the authors of the original paper said that “it is not straightforward to establish a single figure of merit about the stress condition as seen by Raman spectra”. So, I do not consider it a relevant figure for a review.

5- Section 3: Figure 3 is not of a diamond film. So, I find it inappropriate.

6- Section 3: Figure 5 is refereed to a Fe-Cr-Al substrate. But in Figure 5(b) is written “steel”???

7- Section 3: in subsection 3.3, regarding Figure 5, it is stated that “…the adhesion of the diamond film to the substrate appears to be strong”. This cannot be inferred from an indentation with a load of 9.8N. Usually, loads up to 1500N are used; see for example Wei et al, Journal of Crystal Growth 336 (2011) 72–76.

In resume, the manuscript needs a huge revision before resubmission.

Author Response

We thank a lot for the valuable comments. The manuscript has been revised based on the suggestions. The following are our point-to-point responses to all the comments, and responses are in blue.

Responses to Reviewer 3

  1. The present work aims to present a review on the subject of “CVD diamond deposition on iron and steel”, as stated in the title. This is a first problem: nobody does “diamond deposition on iron”. What is technological relevant is “diamond deposition on steel”. Fundamental studies using metallic iron or iron alloys could be important to understand the phenomena when steel is used as substrate but are secondary. So, the title needs revision.

Response: In the review paper, most of the researchers are do jobs about diamond deposition on steel substrates. But there are some jobs indeed containing both the steel and iron substrates, such as refer 20, 37, 48, 52 and so on. As the steels are attractive for most of people, and the iron alloys can be also called as steel, we propose that the title is proper.

  1. As authors know, the goal of “CVD diamond deposition on steel” is to obtain: (1) dense, (2) uniform, (3) adherent CVD diamond coatings, with (4) good quality diamond. So, a review on this subject should present and discuss the main strategies to achieve that. And this should be tentatively done in comparative tables rather than by sequential resumes of papers.

Response: A comparative table may be more useful. But the criterion for different papers has some difference. It is a big challenge for us to make a suitable table.

  1. I liked the organization of the manuscript: starting by the graphitization problem in section 2, followed by section 3 with different approaches to hinder the phenomenon, appearing as subsections 3.1. to 3.5. However, I found the last one (“3.5….DLC films”) completely outside the subject because DLC has nothing to do with diamond and a review on “DLC on steel substrates” would be another manuscript. This is confirmed if you do a search in Web of Science for TITLE (dlc) AND TITLE (steel) where you find 202 papers. So I suggest to remove subsection 3.5.

Response: The section in 3.5 is discussed about DLC films on steel substrates. Indeed, they have some difference to diamond films. But we think that the idea of DLC films is sprung from CVD diamond. Both the diamond and the DLC are aimed to obtain sp3 bonding carbon. Importantly, they provide one way to solve the problem of CVD diamond film. Therefore, we suggest remaining this section in the revised paper.

  1. Another major fault is the exaggerated emphasis the authors put in their previous work, exactly in what is not so relevant: the mechanism of diamond nucleation & growth on a Fe-Cr-Al ternary alloy. Figures 5, 6, 7, 9 and 10 refer to previous results of the authors and are excessive for a balanced presentation of the subjects. It doesn't make sense, for example, that section “4. Diamond nucleation on steel” is based on a paper about nucleation on Fe-Cr-Al.

Response: Indeed, we have done a long, continuing and relevant work about diamond film deposition on steel substrates, especially, about the approaches of alloying elements. The diamond nucleation is an interesting problem. Through our microstructures study of the diamond film delaminated from iron substrate, many unique diamond clusters distributed among the transitional zone are observed. This finding is interesting, as in the previous studies, it was thought that diamond directly nucleated and grew on the catalytic graphite. Additionally, except our own jobs, other related job and some new insights obtained recently have been also discussed in the review paper.  

Other amendments are suggested in the following:

  1. The manuscript requires significant proof reading and revision to improve the quality of English.

Response: We have revised the WHOLE manuscript carefully. And we have tried to avoid any grammar or syntax error in the revised paper.

  1. Section 2: graphite formation mechanism is well known and authors cannot state that they want to propose a new one. Refs [14] and [39] also discuss the question of Fe3C formation as also does: Polini et al, Thin Solid Films 494 (2006) 116-122.

Response: Although it is well known for the catalytic graphite, it is still worthwhile to review the graphite formation mechanism. That is because this is the main drawback for CVD diamond films on steel substrates. Our jobs provide more detailed interfacial investigation by cross-section TEM, and these experiments results may be more direct to understand the mechanism of the catalytic graphite. 

  1. Section 2: Figures 1(b) and (c) are very similar and I don’t see a “film” in figure 1(b). Where is the Fe3C layer in Figures 1(b) or 1(c)?

Response: In TEM image mode, it is indeed difficult to clear present the Fe3C layer due to the thickness influence. In the job of refer 20, there are one HAADF image obtained in the STEM image mode, in which the Fe3C layer with about 2μm thickness located under the catalytic graphite film can be clearly seen.

  1. Section 3: regarding Figure 2, the authors of the original paper said that “it is not straightforward to establish a single figure of merit about the stress condition as seen by Raman spectra”. So, I do not consider it a relevant figure for a review.

Response: We proposed the analysis of the Raman spectra by Damm and Contin et al. provide a way to recognize the interfacial stress condition. Maybe it is not very straightforward, but indeed in many jobs of CVD diamond films, the stresses have been estimated through Raman spectra.

  1. Section 3: Figure 3 is not of a diamond film. So, I find it inappropriate.

Response: We think that after a long deposition, Nakamura et al. (Figure 3 is their job) could also obtain diamond films. Considering their special deposition method, it is worthwhile to refer their picture in the review paper.  

  1. Section 3: Figure 5 is referred to a Fe-Cr-Al substrate. But in Figure 5(b) is written “steel”???

Response: The label of “steel” in Figure 5 has been change to “substrate” in the revised paper. Actually, Fe-Cr-Al alloy is one kind of steels.

  1. Section 3: in subsection 3.3, regarding Figure 5, it is stated that “…the adhesion of the diamond film to the substrate appears to be strong”. This cannot be inferred from an indentation with a load of 9.8N. Usually, loads up to 1500N are used; see for example Wei et al, Journal of Crystal Growth 336 (2011) 72–76. In resume, the manuscript needs a huge revision before resubmission.

Response: The loads about the indentation must consider the thickness of the diamond films. Wei et al’ (Journal of Crystal Growth 336 (2011) 72–76) obtained a relatively thick diamond film, Hence the huge loads would not touch the underneath substrate. But when the diamond film is relatively thin (such as only several micrometers), the influence of substrate under the diamond film must be considered. Additionally, in our job of refer 49 (Thin Solid Films 2008, 516, 3089-3093), a much huge loads of 588N are used.

Reviewer 4 Report

The review is devoted to important problem related creation of diamond coatings on steel/iron surface. Despite some disputable statements and interpretations presented in the manuscript it looks as rather comprehensive and informative overview which may be interesting and helpful for readers.

Author Response

We thank a lot for the valuable comments. The manuscript has been revised based on the suggestions. The following are our point-to-point responses to all the comments, and responses are in blue.

Responses to Reviewer 4

The review is devoted to important problem related creation of diamond coatings on steel/iron surface. Despite some disputable statements and interpretations presented in the manuscript it looks as rather comprehensive and informative overview which may be interesting and helpful for readers.

Response: We have revised the WHOLE manuscript carefully. Some statements have been improved in the revised paper.

Round 2

Reviewer 1 Report

Thank you for all my comments and corrections

Reviewer 3 Report

The authors improved their level of English. Regarding conceptual issues, they kept the work as presented in the first version.